# DiverseFlow: Sample-Efficient Diverse Mode Coverage in Flows

## Abstract

Many real-world applications of flow generative models desire a diverse set of samples covering multiple modes of the target distribution. However, the predominant approach for obtaining diverse sets is not sample-efficient, as it involves independently obtaining many samples from the source distribution and mapping them through the flow until the desired mode coverage is achieved. As an alternative to repeated sampling, we introduce DiverseFlow—a training-free, inference-time approach to improve the diversity of flow models. Our key idea is to employ a determinantal point process to induce a coupling between the samples and drive sample diversity under a fixed sampling budget. We demonstrate the efficacy of DiverseFlow for tasks where sample efficient diversity is highly desirable—text-guided image generation with polysemous words, inverse problems like large-hole inpainting, and class-conditional image synthesis.

**Prompt: "A famous boxer"**     **Prompt: "A letter"**

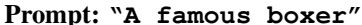
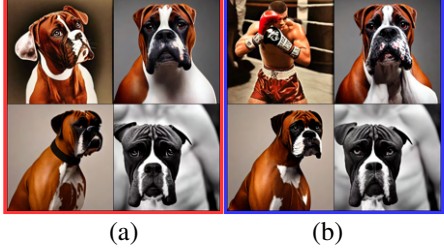
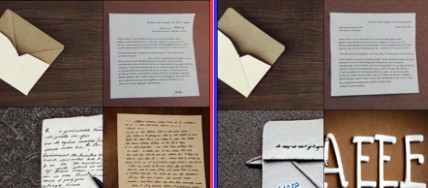

(a)          (b)          (c)          (d)

Figure 1: Examples of text-guided generation with polysemous words. Under a limited sampling budget, regular IID sampling (a, c) may not generate images spanning the different semantic meanings of the words in the prompt. Under the same sampling budget, DiverseFlow (b, d) enhances the diversity of the generated samples and spans different semantic meanings.

## 1 Introduction

Consider the task of text-guided image generation from open-ended prompts, like *"A famous boxer"* or *"A letter"*. Here, the word "boxer" can either mean an *athlete* or a particular *dog breed*. Similarly, the word "letter" may refer to either an *alphabet* symbol or *written correspondence*. If we obtain a few samples from a generative ordinary differential equation (ODE) for each prompt, we observe images depicting only the dog breed and penned correspondences in Figure 1(a) and (c) respectively. This situation necessitates obtaining additional samples from the model, till the desired alternate meanings are discovered. But instead of repeated sampling, can we directly observe more meanings by finding a more *diverse* set?

Beyond the aforementioned examples of text-to-image generation from polysemous[1] prompts, sample diversity is a desirable objective for many other tasks that use generative models. These include inverse problems (e.g., large hole filling) and class-conditioned image generation, to name a few. Diversity or mode coverage is a key pillar in the *generative learning trilemma* (Xiao et al., 2022), in addition to fidelity and latency. For state-of-the-art generative methods such as flow matching models (FM) (Lipman et al., 2022; Liu et al., 2022) and diffusion models (DM) (Sohl-Dickstein et al., 2015;

---

[1]Words or phrases with several meanings.

Ho et al., 2020), significant work has been done on improving the photorealism of samples and the efficiency of the sampling process (Ho & Salimans, 2022; Karras et al., 2022; Lipman et al., 2022; Zheng et al., 2023; Song et al., 2020a; Tong et al., 2023). However, relatively little attention has been paid to explicitly enhancing the diversity of generated samples under a limited sampling budget.

The standard approach to generate a diverse set of images is to repeatedly obtain independent and identically distributed (IID) samples from an easy-to-sample source distribution (e.g., Gaussian distribution), map them to samples in the target distribution, and continue this process until we observe sufficient mode coverage in the target distribution. This process, while effective, is *sample inefficient*, requiring the generation of more images than necessary. Importantly, the mapping from the source to the target density does not hold a linear relationship; even specifically selecting diverse samples from the source distribution by design does not necessarily yield diverse samples in the target distribution. These limitations naturally raise the following research question.

> *How can we generate diverse samples from the target density under a fixed sampling budget?*

In this paper, we propose DiverseFlow to obtain a diverse set of samples in a desired target density under a fixed sampling budget. We focus on deterministic ODE sampling in continuous-time generative models, specifically FMs, an emerging generative paradigm that enables simulation-free training of continuous normalizing flows (CNFs) and includes diffusion as a special case.

DiverseFlow measures the diversity of a set of samples through the *volume* they span in the target space. A set of similar samples span a lower volume, while a diverse set naturally spans a larger volume. We impose a volume-based gradient constraint on the flow ODE by drawing on determinantal point processes (DPP) (Macchi, 1975; Kulesza et al., 2012), a probabilistic model arising from quantum physics that exactly describes the Pauli exclusion principle: that no two fermions may occupy the same quantum state. Figure 1(b,d) show the images generated by DiverseFlow for the prompts "A famous boxer" and "A letter". Unlike the ones generated via IID sampling, those from DiverseFlow span more diverse modes corresponding to the polysemous words in the prompts.

We empirically demonstrate the utility of DiverseFlow across several tasks where diversity is inherently desirable. First, we use DiverseFlow to perform **text-guided image synthesis** for words and phrases that may carry a variety of meanings. Second, we perform **large-hole face inpainting** with occlusion masks covering significant regions of the face that may be important to the person's identity. Third, we apply DiverseFlow on **class-conditioned image synthesis** and demonstrate that we can more efficiently explore the data space compared to IID sampling. Lastly, to better characterize and explain the behavior of DiverseFlow, we perform several experiments on synthetic 2D densities.

**Summary of Contributions**

1. We present a sample-efficient method to obtain a diverse set of samples from a flow ODE (Section 5) and demonstrate it qualitatively (Sections 6.1 to 6.3)
2. We introduce the task of image synthesis from polysemous prompts in the context of analyzing diverse sampling, and show qualitatively and quantitatively that DiverseFlow is able to discover more meanings in Section 6
3. We provide an empirical analysis over various flow matching formulations, showing which are more suitable for obtaining diverse sets (Section 6.4)

## 2 PRELIMINARIES

### 2.1 FLOW MATCHING

Many generative models can be considered as a *transport map* from some easy-to-sample source distribution to an empirically observed yet unknown target distribution. Recent successes in generative modeling represent this transport map in the form of continuous-time processes, such as stochastic differential equations (SDEs) (Song et al., 2020b; Ho et al., 2020), or ordinary differential equations (ODEs) (Lipman et al., 2022; Liu et al., 2022; Albergo et al., 2023). Although diffusion models are formulated as SDEs, a significant body of research focuses on converting the diffusion SDE to a

deterministic ODE at inference time for faster inference. The diffusion ODE, or probability flow ODE, is a particular case of continuous normalizing flows (CNFs). Flow Matching (FM) (Lipman et al., 2022; Liu et al., 2022; Albergo et al., 2023) is motivated by the idea of directly training CNFs in a scalable and simulation-free manner, just like diffusion models. Moreover, many recent text-to-image generative models, such as Stable Diffusion 3 (Esser et al., 2024), adopt the FM framework. As such, we present our approach primarily in the context of FM, and our findings can be extended to diffusion and score-based generative models in a straightforward manner.

A CNF reshapes a prior source density $p_0$ to the empirically observed target density $p_1$ with an ODE of the form:

$$d\mathbf{x}_t = v_\theta(\mathbf{x}_t, t)dt, \quad \mathbf{x}_0 \sim p_0 \tag{1}$$

where $v_\theta$ is a time-dependent velocity field whose parameters $\theta$ are learned; we interchangeably use the notation $v_t$ to imply $v_\theta(\cdot, t)$. It becomes possible to obtain samples from $p_1$ by integrating Equation (1) over time, i.e. by starting at $\mathbf{x}_0 \sim p_0$ for $t = 0$ and solving the ODE till $t = 1$. As our approach is training-free, we do not elaborate on the details of learning to regress the vector field $v_t$; we encourage interested readers to refer to the works of Lipman et al. (2022) and Tong et al. (2023) for a primer on training FMs.

At any timestep $t$ during sampling, an intermediate sample $\mathbf{x}_t$ in the flow trajectory can be used to obtain an approximation of the target as follows:

$$\hat{\mathbf{x}}_1 = \mathbf{x}_t + v_\theta(\mathbf{x}_t, t)(1 - t) \tag{2}$$

Equation (2) is equivalent to simply taking a large Euler step at any time instance $t$ and is naturally more accurate as $t$ approaches $t = 1$. Further, Equation (2) is also well suited for ODEs with 'straight' paths, where the direction of the time-varying velocity $v_t$ remains near-constant in time (such as the work of Liu et al. (2022)). Similarly, we can estimate the source sample by simply taking a step in the reverse direction:

$$\hat{\mathbf{x}}_0 = \mathbf{x}_t - v_\theta(\mathbf{x}_t, t)t \tag{3}$$

## 2.2 DETERMINANTAL POINT PROCESSES

Determinantal point processes (DPPs) (Macchi, 1975; Borodin & Olshanski, 2000; Kulesza et al., 2012) are probabilistic models of repulsion between points. They were originally termed as *fermion processes* as they describe the Pauli exclusion principle or antibunching effect in fermions. To define a DPP, we must first consider a set of points, $\mathcal{Y}$, and a point process $\mathcal{P}(\mathcal{Y})$—a probability measure on $2^{\mathcal{Y}}$ (the set of all possible subsets of $\mathcal{Y}$). $\mathcal{P}$ is *determinantal* when the probability of choosing a random subset $Y \subset \mathcal{Y}$ according to $\mathcal{P}$ is given by:

$$\mathcal{P}(Y \subset \mathcal{Y}) = \frac{\det(\mathbf{L}_Y)}{\sum_{Y \subset \mathcal{Y}} \det(\mathbf{L}_Y)} = \frac{\det(\mathbf{L}_Y)}{\det(\mathbf{L} + \mathbf{I})} \tag{4}$$

where $\mathbf{L} \in \mathbb{R}^{|\mathcal{Y}| \times |\mathcal{Y}|}$ is a kernel matrix, and $\mathbf{L}_Y$ is the sub-kernel matrix indexed by the elements of $Y$. Equation (4) has an intuitive geometric interpretation if we consider the kernel $\mathbf{L}$ to be constructed from cosine similarity: the determinant of $\mathbf{L}_Y$ is the Gram-determinant, describing the squared volume of the $N$-dimensional parallelotope spanned by the set of vectors $Y$. Thus, a DPP naturally assigns higher probabilities to more orthogonal (and thus diverse) subsets that span larger volumes. We leverage DPPs to define a coupled likelihood measure over a set of samples in a flow trajectory.

## 3 RELATED WORK

Efficiently finding diverse sets is useful in several application areas of machine learning. For instance, Batra et al. (2012) show that the M-Best MAP (maximum a posteriori) solutions in Markov random fields are often distant from the ground truth and highly similar. They thus propose the *Diverse M-Best* problem—finding a set of $M$ highly probable solutions satisfying some minimum dissimilarity threshold—that partly inspires our study in Section 4. Yuan & Kitani (2019) utilize DPPs in conjunction with variational autoencoders (VAE) for diverse trajectory forecasting; a set of diverse future pedestrian trajectories improves safety-critical perception systems in autonomous vehicles. Motivated by potential drug discovery and material design applications, Jain et al. (2023)

propose finding diverse Pareto-optimal candidates in a multi-objective setting with generative flow networks.

The work by Corso et al. (2023) which explores diverse non-IID sampling for diffusion models is most similar in spirit to DiverseFlow. However, DiverseFlow is notably different in multiple aspects: 1) Our diversity objective is derived from determinantal point processes, a diversity-promoting probability measure of the joint occurrence of a set of samples. Corso et al. (2023) is instead inspired by stein variational gradient descent (SVGD) (Liu & Wang, 2016). 2) The diversity metric in DiverseFlow (volume, or determinant of similarity kernel) assigns a likelihood score of 0 if any duplicate elements are present; presence of duplicates is tolerated in the Particle Guidance metric (row-wise sum of similarity kernel) 3) DiverseFlow is motivated by imparting diversity to deterministic flows, which lack the inherent stochasticity afforded by SDE formulations of diffusion models that Corso et al. (2023) focuses on.

# 4 DIVERSE SOURCE SAMPLES DO NOT YIELD DIVERSE TARGET SAMPLES

**Problem Setting:** We start with a synthetic example to illustrate our problem of interest. Consider that we have empirical observations from a target distribution $\pi_1 \in \mathbb{R}^2$, which is a random mixture of Gaussians, such as the example shown in Figure 2a. We design $\pi_1 = \sum_{i=1}^{N} w_i \mathcal{N}(\boldsymbol{\mu}_i, \sigma_i^2 \boldsymbol{I})$ to contain $N = 10$ randomly selected modes $\mathcal{N}(\boldsymbol{\mu}_i, \sigma_i^2 \boldsymbol{I})$, each with a random mixture weight $w_i$; we observe that in our example, there are 6 high probability modes and 4 low probability ones. Suppose we have a sampling budget of $K$ samples. This leads to three possible scenarios: (i) $K < N$, (ii) $K = N$, and (iii) $K > N$. Among the aforementioned, case (i) (fewer samples than modes) is the most likely characteristic of any real-world dataset.

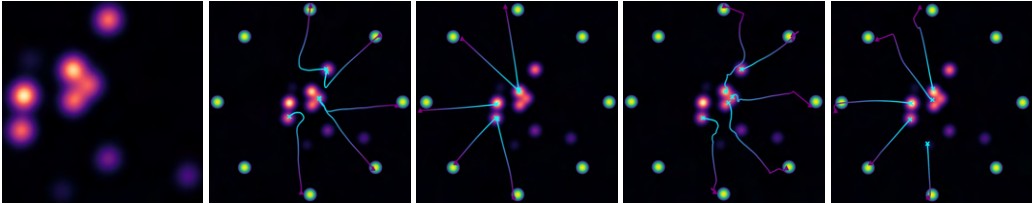

(a) Target Gaussian mixture model density with $N = 10$ modes.
(b) Conditional Flow Matching (Lipman et al., 2022)
(c) Mini-batch Optimal Transport (Tong et al., 2023)
(d) DiverseFlow on CFM
(e) DiverseFlow on Mini-batch OT CFM

Figure 2: We want to find $K = 5$ diverse samples from the target distribution (a) with $N = 10$ modes. Even if samples in the source distribution are diverse, they will not necessarily lead to diverse samples in the target distribution. Even with 5 samples, only three modes are found by IID sampling (b, c). We can find additional modes with the same sampling budget by applying DiverseFlow (d, e).

Let us have a prior distribution $\pi_0$ and some generative model $\Psi$, such that, in the limit of infinite samples, $\Psi(x_0 \sim \pi_0) \sim \pi_1$. Then, the objective of *sample-efficient diverse sampling* is to obtain samples from $\min(K, N)$ modes from $\pi_1$, given a fixed set of $K$ samples in $\pi_0$.

If diverse samples are desired from the target density of the flow, one may make the elementary assumption that *if the particles are distant at the source distribution, after being transported by the flow, they remain distant in the target distribution*. This assumption is not necessarily true, as we show in Figure 2. By design, we choose a uniform mixture of eight Gaussians as the source $\pi_0$ to obtain diverse source samples. In Figure 2b, we can observe that source points from distinct modes can still converge to the same target mode with IID sampling. Thus, an alternative procedure is necessary to obtain a diverse set from a flow in a sample-efficient manner. We further explore this toy problem in Section 6.4.

## 5 DIVERSE SAMPLING FROM FLOWS

From Figure 2, we observe that independently (or heuristically) chosen source samples may not map to a diverse set of target samples. In this case, we can select a new set of source samples and repeat the sampling process till eventually covering at least $K$ modes. However, this approach does not satisfy our fixed sampling budget constraint. An alternative solution to repeated independent sampling is defining and leveraging a diversity measure of the target samples to drive sample diversity. For the set of source samples $\{\mathbf{x}_0^{(1)}, \mathbf{x}_0^{(2)}, \ldots, \mathbf{x}_0^{(k)}\}$, we could optimize a set of perturbations $\{\boldsymbol{\delta}^{(1)}, \boldsymbol{\delta}^{(2)}, \ldots, \boldsymbol{\delta}^{(k)}\}$ such that the new set $\{\mathbf{x}_0^{(1)} + \boldsymbol{\delta}^{(1)}, \mathbf{x}_0^{(2)} + \boldsymbol{\delta}^{(2)}, \ldots, \mathbf{x}_0^{(k)} + \boldsymbol{\delta}^{(k)}\}$ maps to a diverse set of target particles. However, this approach would require multiple simulations of the whole ODE and backpropagating over all the timesteps, which increases the computational complexity of the sampling process over the standard IID sampling.

This leads us to our proposed approach: we avoid multiple simulations and instead optimize the flow trajectory for diversity *while solving the ODE*. For any sample in the flow trajectory $\mathbf{x}_t$, suppose we have an estimate of the target sample $\hat{\mathbf{x}}_1$ through Equation (2). Given a differentiable objective $\mathcal{L}(\{\hat{\mathbf{x}}_1^{(1)}, \hat{\mathbf{x}}_1^{(2)}, \ldots, \hat{\mathbf{x}}_1^{(k)}\})$ that assigns a probability to the diversity of the joint outcome $\{\hat{\mathbf{x}}_1^{(1)}, \hat{\mathbf{x}}_1^{(2)}, \ldots, \hat{\mathbf{x}}_1^{(k)}\}$, it can be leveraged to drive diversity among the target samples by modifying the flow velocity of the $i$-th particle as,

$$\tilde{v}_t^{(i)} = v_t^{(i)} - \gamma(t)\nabla_{\mathbf{x}_t^{(i)}}\mathcal{L}(\{\hat{\mathbf{x}}_1^{(1)}, \hat{\mathbf{x}}_1^{(2)}, \ldots, \hat{\mathbf{x}}_1^{(k)}\}) \tag{5}$$

where $\gamma(t) \in [0, \infty)$ is a time-varying scale that controls the strength of the diversity gradient. Setting $\gamma(t) = 0$ reduces to the standard IID sampling scenario, while $\gamma(t) > 0$ will encourage diversity between the generated samples. In practice, $\gamma(t)$ follows the schedule of the probability path normalized by the norm of the DPP gradient.

### 5.1 DETERMINANTAL GRADIENT CONSTRAINTS

We desire objective $\mathcal{L}$ in Equation (5) to be higher if the items in the set are diverse and lower if they are similar to each other. We interpret diversity in terms of the *volume* spanned by the set. Consider that we have $k$ samples in $\mathbb{R}^d$ (assume $k < d$). An objective that prefers diversity can be defined as the volume of the $k$-dimensional parallelotope in $\mathbb{R}^d$ spanned by the sample vectors; this volume becomes diminished when there are similar samples (and even zero, for identical samples). The determinant describes volumes well; a diverse set must span a large volume in the sample space and have a corresponding large determinant.

To define a measure over a set of samples, we draw on the idea of determinantal point processes (DPP). We first define a kernel $\mathbf{L}(\{\hat{\mathbf{x}}_1^{(1)}, \hat{\mathbf{x}}_1^{(2)}, \ldots, \hat{\mathbf{x}}_1^{(k)}\})$ as follows:

$$\mathbf{L}^{(ij)} = \exp\left(-h\frac{\|\hat{\mathbf{x}}_1^{(i)} - \hat{\mathbf{x}}_1^{(j)}\|_2^2}{\text{med}(\mathbf{U}(\mathbf{D}))}\right) \tag{6}$$

where $\mathbf{D}$ denotes a distance matrix with $\mathbf{D}_{ij} = \|\mathbf{x}^{(i)} - \mathbf{x}^{(j)}\|_2^2$, $\mathbf{U}(\mathbf{D})$ denotes the upper triangle entries of $\mathbf{D}$, $h$ denotes a kernel spread parameter, and $\text{med}(\mathbf{U}(\mathbf{D}))$ denotes the median of those entries. Given $\mathbf{L}$, we may define a DPP-based likelihood as:

$$\mathcal{L}(\{\hat{\mathbf{x}}_1^{(1)}, \hat{\mathbf{x}}_1^{(2)}, \ldots, \hat{\mathbf{x}}_1^{(k)}\}) = \frac{\det(\mathbf{L})}{\det(\mathbf{L} + \mathbf{I})} = \prod_{a=1}^{k} \frac{\lambda(\mathbf{L})_a}{1 + \lambda(\mathbf{L})_a} \tag{7}$$

where $\lambda(\mathbf{L})_a$ is the $a^{\text{th}}$ eigenvalue of the kernel $\mathbf{L}$. The log-likelihood is then,

$$\mathcal{LL} = \log\det(\mathbf{L}) - \log\det(\mathbf{L} + \mathbf{I}) \tag{8}$$

Note that the Euclidean distance $\|\hat{\mathbf{x}}_1^{(i)} - \hat{\mathbf{x}}_1^{(j)}\|_2^2$ is not very meaningful in the high-dimensional raw image space (Aggarwal et al., 2001). Therefore, in practice, the distance should be computed in a robust feature space, i.e., $\|F(\hat{\mathbf{x}}_1^{(i)}) - F(\hat{\mathbf{x}}_1^{(j)})\|_2^2$, where $F$ is some domain-specific feature extractor, such as the vision transformer (ViT) (Dosovitskiy et al., 2020) for images.

**Quality Constraint:** The DPP defined in Equation (7) acts as a repulsion-seeking force on the flow ODE. A quality term can be incorporated into the DPP kernel to regularize the trajectory diversification. Although flows can be defined between any arbitrary two distributions, let us consider the special case when the source is a Gaussian, i.e., $p_0 \sim \mathcal{N}(0, \mathbf{I})$. Suppose we have a quality vector $\mathbf{q}_t = \{q^{(1)}(t), q^{(2)}(t), \ldots, q^{(k)}(t)\}$, where any $q^{(i)}(t) \in [0, 1]$. We can then define a new kernel $\mathbf{L}_q = \mathbf{L} \odot \mathbf{q}_t \mathbf{q}_t^T$, where each $q^{(i)}(t)$ penalizes a sample $\mathbf{x}_t^{(i)}$ if it deviates too much from the flow. To define this, we obtain an estimate of the source sample $\hat{\mathbf{x}}_0^{(i)}(t)$ for any given sample $\mathbf{x}_t^{(i)}$ via Equation (3), and check if it lies within a desired percentile-radius $\rho$ of the Gaussian $p_0$. Specifically, we define the time-dependent sample quality as

$$
q^{(i)}(t) = \begin{cases} 1 & \text{if } \|\hat{\mathbf{x}}_0^{(i)}(t)\|_2^2 \leq \rho^2 \\ \max\left(\epsilon, e^{-\left(\|\hat{\mathbf{x}}_0^{(i)}(t)\|_2^2 - \rho^2\right)}\right) & \text{otherwise} \end{cases} \tag{9}
$$

where $\epsilon$ is a 'minimum quality' we assign to prevent a zero determinant.

**Soft DPP Objective:** Note that the exact log-likelihood can still be undefined on the rare occasion when we have very similar elements in the set. Instead of maximizing $\sum_a \log(\lambda_a/(1 + \lambda_a))$ we can maximize the expectation of the cardinality of the DPP (or the approximate rank of $\mathbf{L}$):

$$
\mathbb{E}\left[ |\ \{\hat{\mathbf{x}}_1^{(1)}, \hat{\mathbf{x}}_1^{(2)}, \ldots, \hat{\mathbf{x}}_1^{(k)}\}\ | \right] = \sum_{a=1}^{k} \frac{\lambda(\mathbf{L})_a}{\lambda(\mathbf{L})_a + 1} = \text{Tr}(\mathbf{I} - (\mathbf{L} + \mathbf{I})^{-1}) \tag{10}
$$

For cases where the DPP volume is not well defined (such as when $n > d$, like on the 2D plane), we adopt Equation (10). In other scenarios (such as high dimensional examples in Section 6.1) we use the exact log-likelihood $\mathcal{LL}$ defined in Equation (8).

## 5.2 Coupled Ordinary Differential Equations

At any timestep $t$, the measure of diversity in Equation (8) or Equation (10) can be adopted to modify the flow of the $i$-th particle. We compute the gradient of the samples with respect to the diversity measure and use it to modify the ODE as follows:

$$
d\mathbf{x}_t^{(i)} = \left[ v_\theta(\mathbf{x}_t^{(i)}, t) - \gamma(t) \nabla_{\mathbf{x}_t^{(i)}} \log \mathcal{L}(\{\hat{\mathbf{x}}_1^{(1)}, \hat{\mathbf{x}}_1^{(2)}, \ldots, \hat{\mathbf{x}}_1^{(k)}\}) \right] dt \tag{11}
$$

Where $\gamma(t)$ is a time-varying scaling factor. Unlike the IID sampling scenario where we have $K$ independent ODEs, Equation (11) corresponds to a system of coupled non-linear ordinary differential ordinary equations. To see this, first note that the estimate $\hat{\mathbf{x}}_1^{(i)}$ depends on the current sample $\mathbf{x}_t^{(i)}$ i.e., $\hat{\mathbf{x}}_1^{(i)} = \mathbf{x}_t^{(i)} + v_\theta(\mathbf{x}_t^{(i)}, t)(1 - t)$. Second, the DPP log-likelihood $\mathcal{LL}(\{\hat{\mathbf{x}}_1^{(1)}, \hat{\mathbf{x}}_1^{(2)}, \ldots, \hat{\mathbf{x}}_1^{(k)}\})$ induces a time-dependent coupling between the $K$ trajectories of $\mathbf{x}_t^{(i)}, i = 1, \ldots, K$ and seeks to diversify the target samples. Although higher-order ODE solvers (Karras et al., 2022) can be employed to solve the coupled ODEs, we use the standard Euler method.

## 6 Experiments

We demonstrate the utility of DiverseFlow in flow matching models; we consider three applications where sample diversity is naturally desirable: text-guided image generation with polysemous words and large-hole inpainting and class-conditional image generation. We also analyze the effect of DiverseFlow on different flow matching formulations w.r.t. its ability to span diverse modes through a synthetically constructed 2D density example.

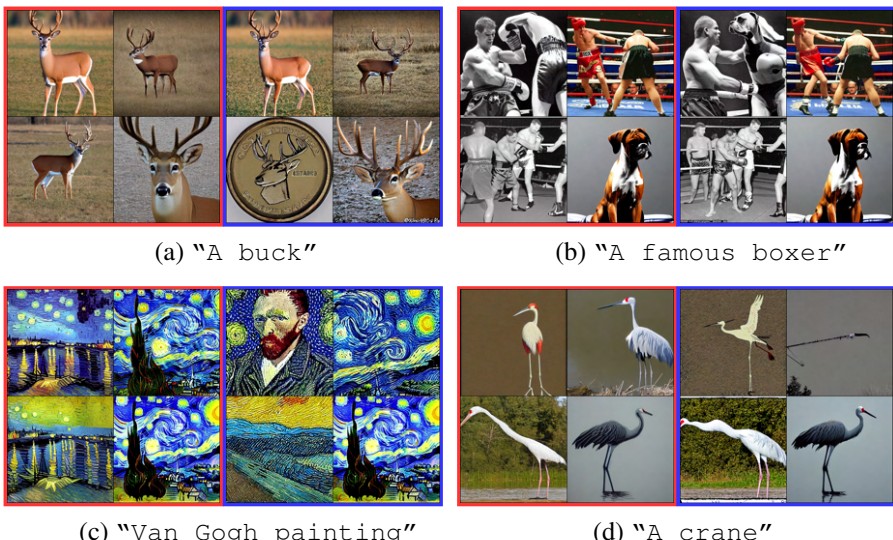

(a) "A buck"  (b) "A famous boxer"

(c) "Van Gogh painting"  (d) "A crane"

Figure 3: We show a set of four prompts that may have multiple meanings. For each prompt, the left image (red box) denotes regular sampling with CFG=8, while the right image (blue box) shows the result after incorporating DiverseFlow. We demonstrate that for the same fixed source, DiverseFlow is often able to find more diverse sets and additional meanings.

## 6.1 Image Generation from Polysemous Prompts

In text-to-image generation, the conditional data distribution corresponding to a text prompt may contain many variations, and it is a desirable objective to generate images that span those variations in a sample-efficient manner. We pose a scenario where diverse sets are easily observable: when an open-ended text prompt is *polysemous* and carries *multiple meanings*, such as the examples we show in Figure 1 and Figure 3. In Figure 3(a), the prompt "A buck" may commonly refer to a male deer. However, it may also informally refer to a United States dollar. Using the *same* four source points, which are deterministically mapped to four deer images by IID sampling, DiverseFlow finds a different set of samples—one that includes a dollar-like coin, albeit embossed with a deer head. We also observe minor differences between the two sets of images, such as changes in pose and background in the top-right and bottom-right deers.

Figure 3(d) finds a crane (a large machine used in construction) from an original set comprised of four birds. In (b), although the standard IID samples depict multiple meanings (dog breed and athlete), three images depict athletes, while only one shows a dog. By improving the diversity of the set, DiverseFlow finds dogs in two images and generates a rare example of *a dog-headed man engaging in boxing* (top-right). For Figure 3(c), while 'Van Gogh painting' is not quite a polysemous word, it can still have two meanings: a painting *painted by* Van Gogh, or a painting *of* Van Gogh. The regular samples contain minimal diversity, as they include two sets of repeated paintings of Van Gogh. With DiverseFlow, not only can we get a set of four distinct paintings, but we also have a portrait of Van Gogh, which is one of the additional meanings of the prompt. However, DiverseFlow is limited by the generative mapping learned by the flow; it is not always possible to discover diverse meanings. We show some additional examples in the appendix, in Figure 10 and Figure 11 respectively.

## 6.2 Diverse Inpainting on Faces

Another inverse problem where diverse solutions are desirable is face inpainting, where we seek to inpaint the missing parts of the face with diverse plausible facial textures and structures. To demonstrate inpainting with FM models, we first incorporate Manifold Constrained Gradient (MCG) (Chung et al., 2022) in an off-the-shelf unconditional Rectified-Flow model. In addition to the manifold constraints, we employ determinantal gradient constraints to enhance diversity. The complete flow-based inpainting method is described in Algorithm 1. In Figure 4 (b), we observe that the inpainted faces of the four women have similar expressions (largely neutral). DiverseFlow

improves the diversity of the set by yielding a highly different expression in the top-right image. In (d) and (e), we also observe changes in facial hair and expressions due to diversification.

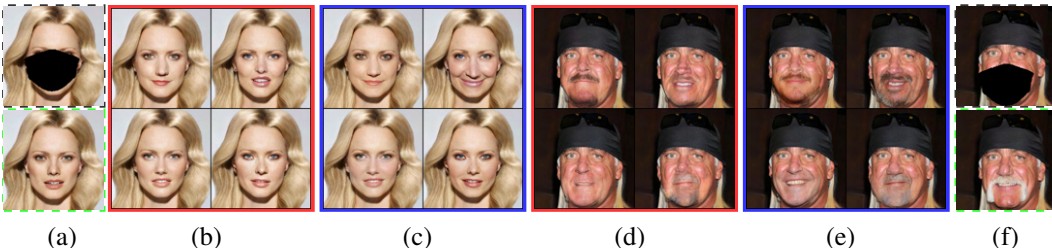

(a)      (b)      (c)      (d)      (e)      (f)

Figure 4: Inpainting on CelebAHQ-256 × 256; (a, f) dashed boxes show masked input (top) and ground truth (bottom) respectively. (b, d) RectifiedFlow (Liu et al., 2022) + MCG (Chung et al., 2022) (c, e) DiverseFlow applied on RectifiedFlow + MCG

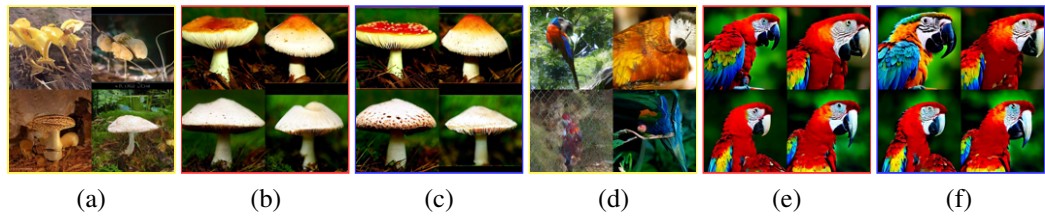

(a)      (b)      (c)      (d)      (e)      (f)

Figure 5: Class-conditional ImageNet samples from LFM (Dao et al., 2023). We show samples for two classes, (a, b, c) 'Mushroom' (class 947) and (d, e, f) 'Macaw' (class 88). (a, d) No CFG. (b, e) LFM with CFG. (d, f) LFM with CFG and DiverseFlow.

## 6.3 DIVERSE CLASS-CONDITIONAL IMAGE SYNTHESIS

Suppose we can access a class-conditioned flow matching (FM) model trained on an unknown image dataset. To explore the *unobservable* true dataset, we may use a set of class-conditional samples from the FM model. We adopt a latent flow matching (LFM) model (Dao et al., 2023), trained to generate 256 × 256 resolution images from the ImageNet (Deng et al., 2009) dataset. Much like latent diffusion, LFM employs classifier-free guidance to create high-quality samples. However, this naturally poses a cost to diversity, as we show in Figure 5.

By incorporating DiverseFlow, we can maintain the high quality of the samples and simultaneously explore more modes in the dataset. In Figure 5, we demonstrate two ImageNet classes that may have diversity: 'Mushroom' and 'Macaw.' For mushrooms, we observe that LFM primarily generates two species of mushrooms. However, by applying DiverseFlow, we successfully find a new species within our limited set: an *Amanita muscaria*, also known as the *fly agaric*—easily distinguishable by the white spots on its red cap. In another example, we see that while LFM generates four scarlet macaws, using the same source samples, DiverseFlow helps us find a different blue and yellow macaw. In all samples shown in Figure 5, we use 100 Euler steps. For classifier-free guidance, we use a guidance strength of 8. Additional details are provided in the Appendix.

## 6.4 DIVERSEFLOW ACROSS VARIOUS FM FORMULATIONS

In Figure 6, we study the utility of DiverseFlow across four different flow matching (FM) formulations and observe two common properties: (i) DiverseFlow does not have a large effect on target samples that are already diverse (ii) FM models tend to map near-identical source samples to highly similar target samples. This phenomenon can be overcome with DiverseFlow. Additionally, we observe that the trajectories of the non-diversified samples remain largely unchanged in OT-CFM (b) and SB-CFM (c). In contrast, we observe additional curving in CFM (a) and SI (d). We hypothesize that this is because estimates of $\hat{x}_1$ in CFM and SI have inaccuracies, leading to the flow direction changing significantly with time. We also perform a numerical experiment to quantify the average number of modes discovered by each FM variant on the toy density we define in Figure 2a as the sampling budget $K$ increases. Figure 7 reports the results. For a maximum sampling budget of 10, OT-CFM discovers only 5.64 modes on average, which is expected since the dataset contains

six high-probability modes. By incorporating DiverseFlow, we can find 7.11 modes on average. We observe that OT-CFM and SB-CFM benefit most from DiverseFlow, which conforms with our previous hypothesis that straight paths are beneficial for accurately estimating the diversity gradient.

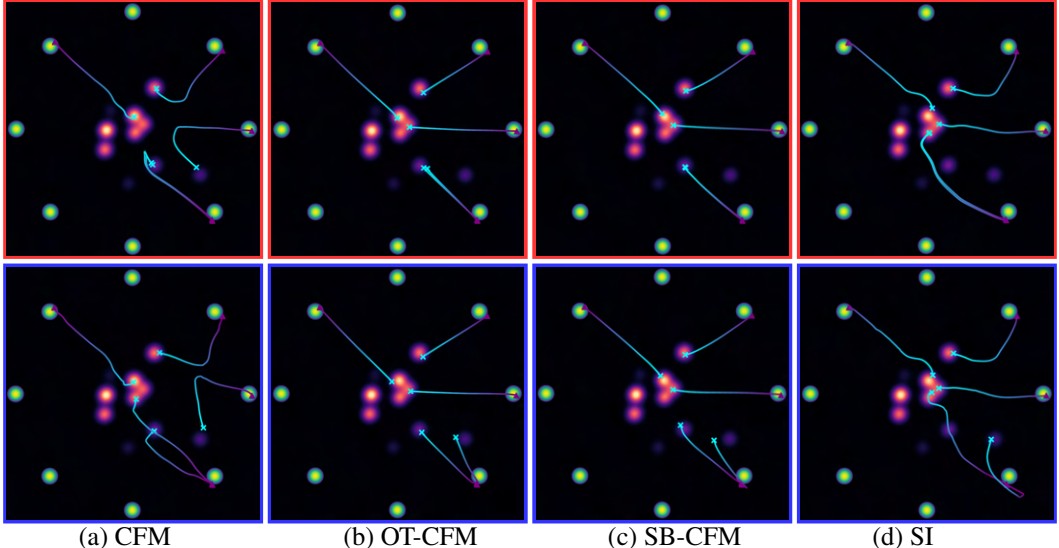

(a) CFM      (b) OT-CFM      (c) SB-CFM      (d) SI

Figure 6: In IID sampling (top row), a pair near-identical source samples result in nearly identical target samples. DiverseFlow (bottom row) forces the similar source samples apart in the flow trajectory, and finds distinct modes.

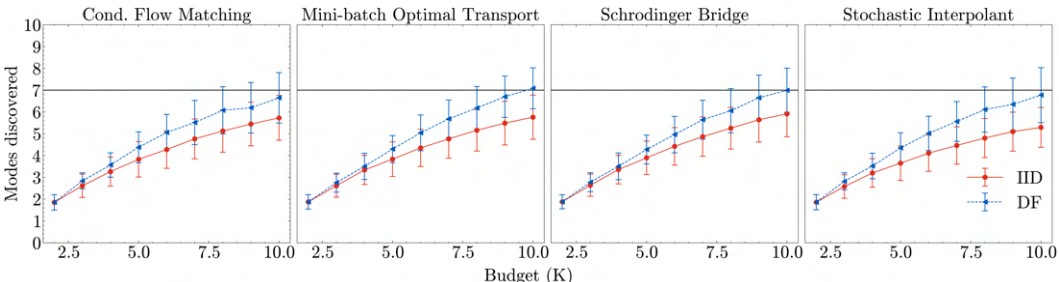

Figure 7: Comparing different FM formulations in terms of the number of modes spanned by IID sampling versus with DiverseFlow. More details about the experiment are provided in the Appendix.

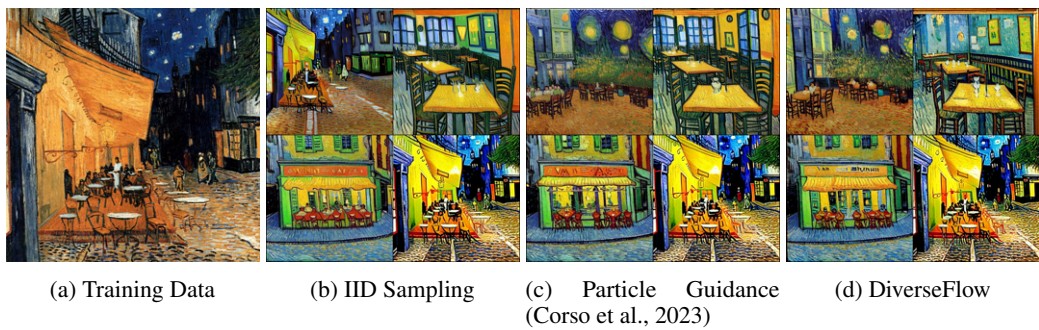

(a) Training Data      (b) IID Sampling      (c) Particle Guidance (Corso et al., 2023)      (d) DiverseFlow

Figure 8: An example adopted from Corso et al. (2023): Particle Guidance (c) and DiverseFlow (d) for the prompt "VAN GOGH CAFE TERASSE copy.jpg"; the original data is shown in (a).

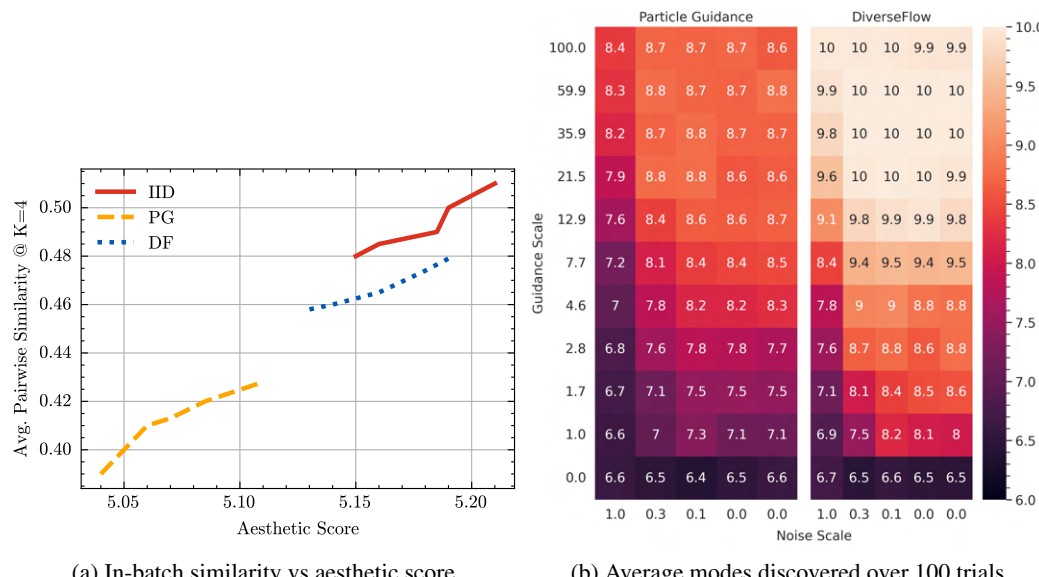

(a) In-batch similarity vs aesthetic score  (b) Average modes discovered over 100 trials

Figure 9: Comparing DiverseFlow and Particle GUidance

## 6.5 COMPARISON TO PARTICLE GUIDANCE

Previously, Corso et al. (2023) demonstrate that their method can alleviate Stable Diffusion's training data regurgitation problem (Somepalli et al., 2023) to some extent. In Figure 8, we demonstrate similar capabilities; the top left and bottom right examples are copies of the training data. Subsequently, both methods find a new example for the top-left sample. In high dimensional data, the number of modes, $N$, is significantly greater than the budget $K$, so finding unique modes is still highly probable. To better highlight the differences between DiverseFlow and Particle Guidance, we adopt another experiment proposed by Corso et al. (2023): finding modes in a *uniform* Mixture of Gaussian distribution. Unlike the asymmetric distribution we utilize in Figure 6, we now have a mixture of $N = 10$ modes, where each mode has an equal probability. Corso et al. (2023) provides the result that IID sampling discovers **about 6.5 modes** on average, while Particle Guidance with a Euclidean kernel discovers **almost 9 out of 10 modes**. We verify this result in Figure 9b, finding that Particle Guidance discovers up to **8.8 modes** (averaged over 100 trials). However, by using DiverseFlow, it is possible to discover **all 10 modes**, showing that our approach has a stronger diversification effect. We also compare the diversity versus quality of DiverseFlow against Particle Guidance over 30 polysemous prompts (repeated over 10 seeds) in Figure 9a. Quality is measured by Aesthetic Score (higher is better) (Christophschuhmann), and diversity is measured by average pairwise similarity of a set (lower is better) (Corso et al., 2023). We observe that though DiverseFlow obtains better diversity at similar quality, the aesthetic score of Particle Guidance is quite low, suggesting poor quality. In Figure 12 we highlight the fact that Particle Guidance can sometimes suffer from artefacts, and does not find as many diverse modes as DiverseFlow.

## 7 CONCLUSION

In numerous generative model applications, generating diverse samples under a fixed sampling budget is a critical requirement. Flow matching is an emerging generative modeling paradigm that alleviates key issues in diffusion and continuous normalizing flow-based generative models. However, the deterministic nature of flow-matching models inherently limits their ability to enhance the diversity of the generated samples in a sample-efficient manner. In this paper, we proposed DiverseFlow to enforce diversity among a set of generated samples by coupling them through a determinantal point process and accounting for the quality of the samples. Across multiple generative applications that inherently desire diverse samples, we demonstrated that DiverseFlow can efficiently enhance the diversity and mode coverage of the samples in the target distribution.

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

## A  COMPUTATIONAL DETAILS

All experiments were performed on a single NVIDIA RTX A6000 GPU with 48 GB memory. All images generated from latent-space models were generated at FP32 precision.

## B  ABOUT EXPERIMENTS

### B.1  POLYSEMOUS PROMPTS

For direct comparison to (Corso et al., 2023), we utilize the probability flow ODE formulation of Stable Diffusion v1.5 (Rombach et al., 2022) with polysemous prompts. We also apply DiverseFlow on the larger Stable Diffusion v3 model (Esser et al., 2024), which is based on rectified flows (Liu et al., 2022). We show some results for SD-v3 in Figure 10.

We adopt 30 polysemous prompts, which are given in Appendix B.1. To find such prompts, we prompted an LLM for 50 polysemous nouns, and then we manually filtered 30 good polysemous words with clearly distinct meanings. We use 30 Euler steps to sample from SD-v1.5, and 28 Euler steps for SD-v3, with a classifier-free guidance strength of 8 and 7 respectively. For the feature extractor, we experiment with both CLIP-ViT-B16 and DINO-ViT-B8, and find better results with DINO. From Appendix B.1, it can be seen that polysemous prompts are a challenging task; for many prompts, it is not yet possible to find the diverse meanings. For example, for "a spring", both SD-v1.5 and SD-v3 only yield images of the season, and not the coiled object. DiverseFlow helps discover 5 and 4 additional meanings for SD-v1.5 and SD-v3 respectively. For the images in Figure 12 and the results in Figure 9a, we use a scaling factor of $8\sigma(t)$ for Particle Guidance, same as used by the authors in their paper. For DiverseFlow, we use $\frac{20\sigma(t)}{\|\nabla \log \mathcal{L}(\mathbf{x_t}^{(1)}, \mathbf{x_t}^{(2)}, ..., \mathbf{x_t}^{(k)})\|}$.

### B.2  INPAINTING

To perform inpainting with an FM model, we first adopt an *unconditional* off-the-shelf face image generating FM. We adopt a RectifiedFlow model pre-trained on CelebAHQ-256 $\times$ 256 (Karras et al., 2018), from `https://github.com/gnobitab/RectifiedFlow`. Next, we extend the manifold constrained gradient (MCG) algorithm (Chung et al., 2022) for diffusion to FM models, in Algorithm 1. We use $\gamma(t) = 10 \frac{\sqrt{1-t}}{\|\nabla \log \mathcal{L}\|}$ as a time-varying scale for the DPP gradient.

The images in Figure 4 were generated with 200 Euler ODE steps; we used the seed 0 across all images. The masks we used for inpainting were arbitrarily chosen to hide large areas of the face and not from any particular dataset.

For the feature encoder $F$, we use the FaRL model (Zheng et al., 2021), which is a CLIP-like model trained on LAIONFace Zheng et al. (2022). FaRL is trained in a mask-aware manner, and we downsample the inpainting mask to additionally create an attention mask, to ensure that the feature encoder $F$ does not focus on the irrelevant areas.

### B.3  CLASS-CONDITIONED IMAGE GENERATION

For the ImageNet samples, we show in Figure 5, we use pre-trained LFM models from: `https://github.com/VinAIResearch/LFM`, specifically the 'imnet_f8_ditb2' weights. The mushroom

| polysemous word | SD-v1.5 | SD-v1.5+DF | SD-v3 | SD-v3 + DF |
|:---:|:---:|:---:|:---:|:---:|
| boxer | ✓ | ✓ | ✗ | ✓ |
| crane | ✓ | ✓ | ✓ | ✓ |
| bat | ✗ | ✗ | ✗ | ✗ |
| letter | ✓ | ✓ | ✓ | ✓ |
| buck | ✓ | ✓ | ✗ | ✗ |
| seal | ✓ | ✓ | ✗ | ✗ |
| mouse | ✗ | ✗ | ✗ | ✗ |
| horn | ✓ | ✓ | ✓ | ✓ |
| chest | ✗ | ✗ | ✗ | ✗ |
| nail | ✓ | ✓ | ✓ | ✓ |
| ruler | ✗ | ✓ | ✗ | ✓ |
| ball | ✗ | ✗ | ✗ | ✗ |
| file | ✓ | ✓ | ✓ | ✓ |
| ring | ✗ | ✗ | ✗ | ✗ |
| deck | ✗ | ✗ | ✗ | ✗ |
| nut | ✗ | ✗ | ✗ | ✗ |
| bolt | ✗ | ✓ | ✓ | ✓ |
| bow | ✗ | ✗ | ✗ | ✗ |
| pupil | ✗ | ✗ | ✗ | ✗ |
| palm | ✗ | ✓ | ✓ | ✓ |
| pitcher | ✗ | ✗ | ✓ | ✓ |
| fan | ✗ | ✓ | ✗ | ✗ |
| club | ✓ | ✓ | ✓ | ✓ |
| anchor | ✗ | ✗ | ✗ | ✗ |
| mint | ✓ | ✓ | ✗ | ✓ |
| iron | ✗ | ✓ | ✗ | ✓ |
| bank | ✗ | ✗ | ✗ | ✗ |
| glass | ✗ | ✗ | ✗ | ✗ |
| pen | ✗ | ✗ | ✗ | ✗ |
| spring | ✗ | ✗ | ✗ | ✗ |
| total | 10 | 15 | 9 | 13 |

Table 1: List of polysemous prompts and discovered diverse meanings.

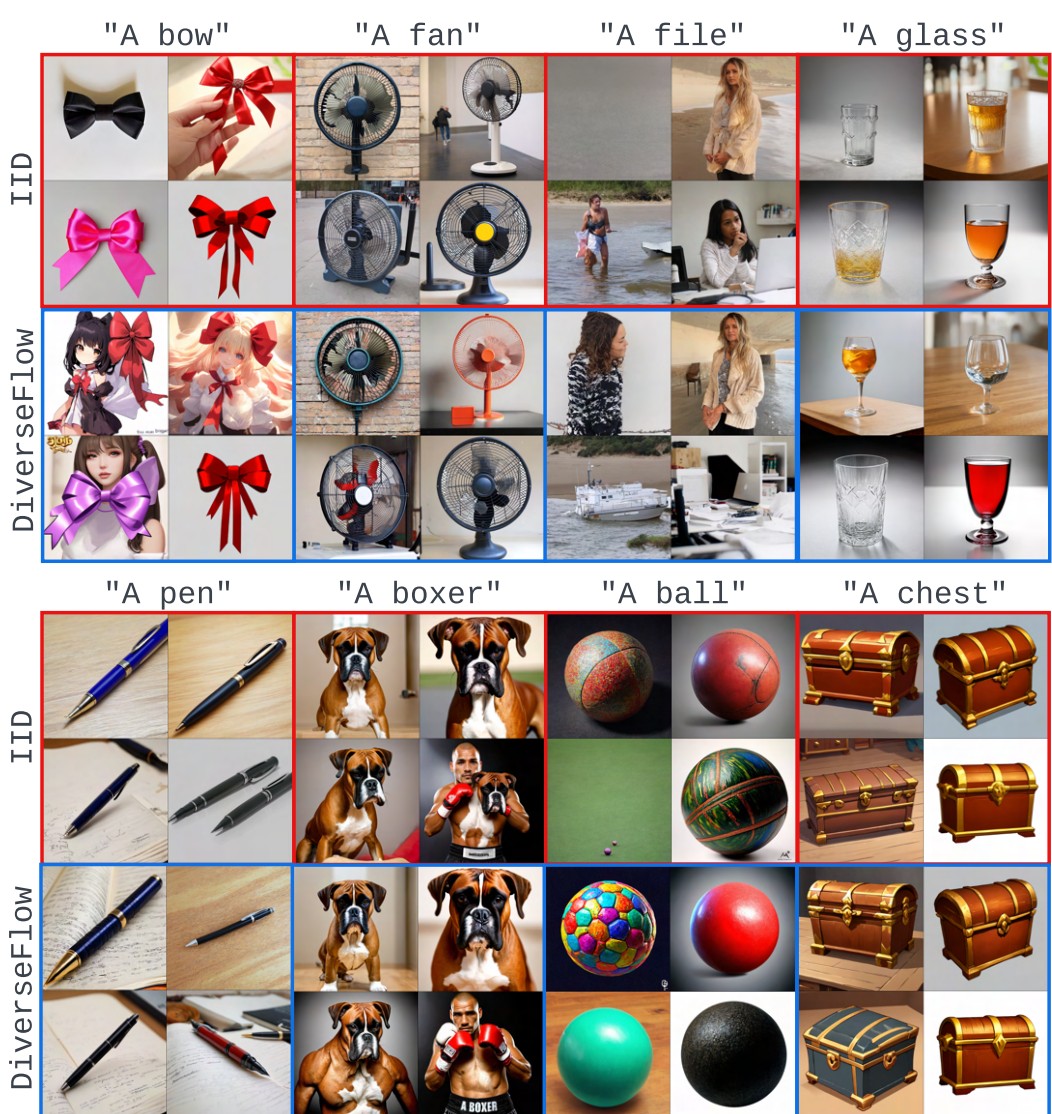

Figure 10: Some examples on SD3 where significantly polysemous meanings are not discovered. However, DiverseFlow still yields more diverse samples compared to IID samples.

(class 947) example was generated with seed 0, while the macaw (class 88) example was generated with seed 6. Additionally, we used 100 Euler steps and a classifier-free guidance strength of 8 for both samples. We primarily used DINO-ViT-B8 as the feature extractor $F$. The reason behind choosing the classes 947 and 88 for our small qualitative example in Figure 4 is that these two classes are prominently featured on the LFM project webpage: `https://vinairesearch.github.io/LFM`.

### B.4 MODE FINDING

We train a set of four identical models from scratch for the four FM variants used in Figure 7. Each model is an *unconditional* generative model and is defined as an MLP consisting of 4 fully connected layers, each except the first having 256 hidden units; the first layer has a hidden size of 256 + 1 to account for the time input. We use the `torchcfm` library (`https://github.com/atong01/conditional-flow-matching`) for the conditional path construction.

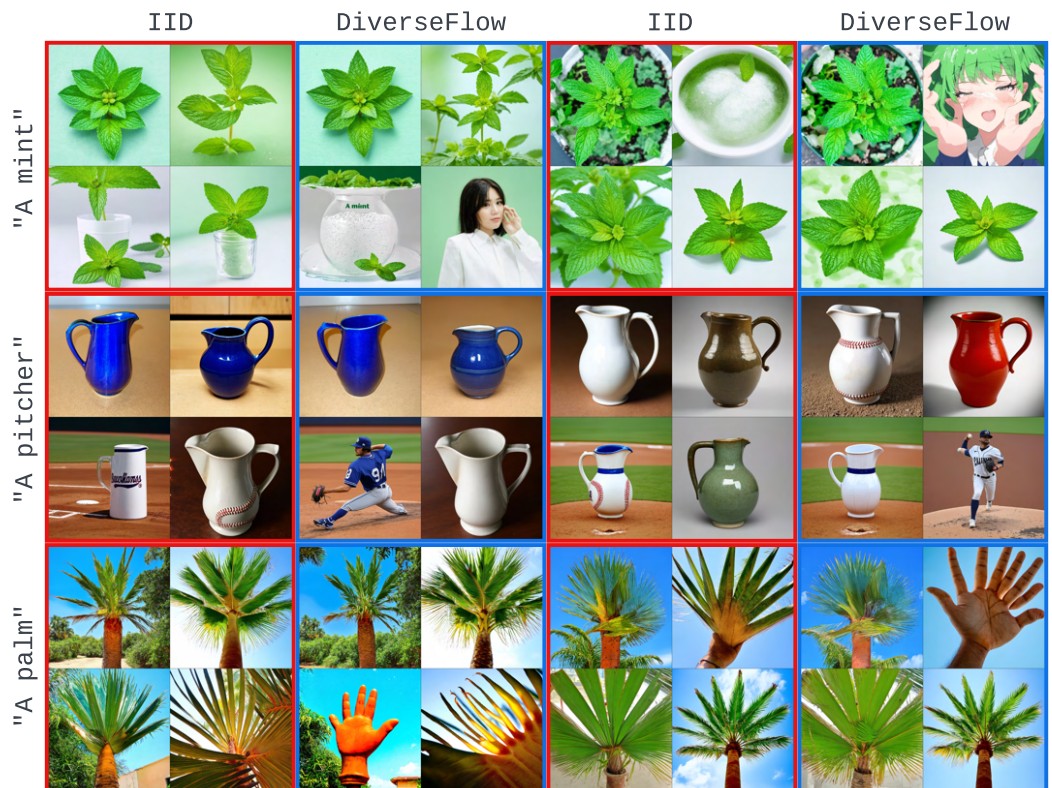

Figure 11: Some examples on SD3 where DiverseFlow discovers alternate meanings that IID sampling doesn't find.

We solve the ODE with an Euler solver with 100 steps. We start with a budget of $K = 2$ (as for $K = 1$, the ODE must always find at least 1 mode) and increase $K$ till $K = N = 10$, where $N = 10$ is the true number of modes in the dataset. For each $K$, we repeat 1000 trials (by taking random seeds 0-999). We use $\gamma(t) = 2\frac{\sqrt{1-t}}{\|\nabla \log \mathcal{L}\|}$. Since the data is 2D, we do not use any feature encoder $F$.

We find $\sim 7$ modes on average with DiverseFlow, while IID sampling finds $\sim 5.6$ modes. With regular IID sampling, the least diverse seems to be the Stochastic Interpolant (Albergo et al., 2023). Additionally, for the quantity 'maximum modes found at any trial' we observe that in over 1000 trials with a budget of $K = 10$, IID sampling does not find a single instance of all 10 modes in any CFM formulation.

### B.5 MODE-FINDING WITH IDEAL SCORE

In Figure 9b, no model is trained, and we have access to a true score function of a mixture of uniform Gaussian distribution, as shown in Figure 13. We scale the DPP gradient by $\gamma(t) = W\frac{\sigma(t)}{\|\nabla \log \mathcal{L}\|}$, where $\sigma(t)$ is the variance schedule path, and $W$ is a variable temperature parameter (Y-axis in Figure 9b).

## C CONNECTIONS TO PARTICLE GUIDANCE

It is possible to formulate Particle Guidance in DiverseFlow's framework. Consider the DPP kernel $\mathbf{L}$ that we define in Equation (6). Particle Guidance defines a time-varying 'log potential' that takes the form:

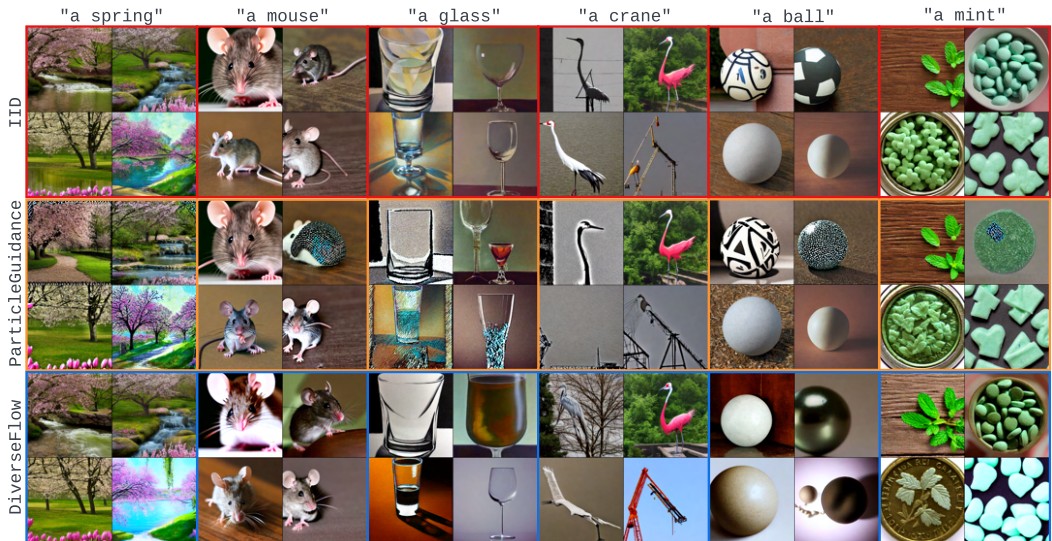

Figure 12: We find that Particle Guidance (middle row) can occasionally suffer from strange artefacts. We hypothesize this may be because of the lack of any regularization in the approach of (Corso et al., 2023), and the problem is particularly highlighted for open-ended prompts. In addition to retaining quality, note that DiverseFlow (bottom row) finds more meanings: for example, for "a mint" (right-most column), DiverseFlow discovered three meanings (the plant, the candy, coinage production) with four samples.

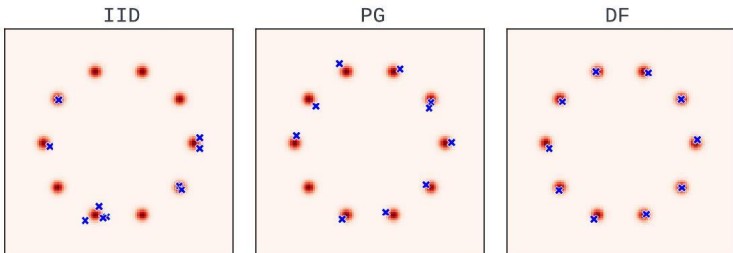

Figure 13: Finding modes on uniform mixture of Gaussian with true score

$$\log \Phi_t^{(i)}(\mathbf{x}^{(1)}, \mathbf{x}^{(2)}, \dots, \mathbf{x}^{(k)}) = \sum_j \mathbf{L}^{(ij)} \qquad (12)$$

That is, the log potential for each particle is its pairwise similarity with every other particle. However, it is not readily apparent why the log potential is this pairwise sum (Equation 4 in particle guidance paper). In our work, the DPP is a probability measure that yields an approximate likelihood of the joint distribution $p(\mathbf{x}^{(1)}, \mathbf{x}^{(2)}, \dots, \mathbf{x}^{(k)})$. Therefore, the log potential is simply the log-likelihood of the DPP. One geometric way to interpret the two approaches may be observed in Figure 14.

Thus, the log potential for each particle in particle guidance is distinct. However in our work, the potential is the same for any particle, as it is defined over the determinant. The kernel-sum utilized in Particle Guidance can also be interpreted as an approximate joint likelihood function, except, unlike the DPP, it assigns a non-zero likelihood to the occurrence of duplicate elements. It is thus a softer form of diversification, which can be observed in Figure 9b. Finally, particle guidance does not consider a quality factor on the kernel, unlike DPP-based methods.

**Algorithm 1** MCG Flow Inpainting with Euler Method

---

**Require:** Inpainting input $\mathbf{Y} \in \mathbb{R}^{H \times W \times 3}$, inpainting mask $\mathbf{M} \in \mathbb{Z}_2^{H \times W \times 3}$, number of sampling steps $N$, time-varying velocity field $v_\theta$

$\quad \mathbf{X}_0 \sim \mathcal{N}(0, \mathbf{I})$            ▷ Sample a particle from source distribution $\mathbf{Z}_0$

$\quad$**for** i=0 … $N - 1$ **do**

$\quad\quad t_i, t_{i+1} \leftarrow \frac{i}{N}, \frac{i+1}{N}$            ▷ Uniform timesteps, $t \in 0 \ldots 1$

$\quad\quad \Delta_t \leftarrow t_{i+1} - t_i$

$\quad\quad \mathbf{V}_i \leftarrow v_\theta(\mathbf{X}_i, t)$            ▷ Predicted velocity at timestep $t$

$\quad\quad \hat{\mathbf{X}}_N \leftarrow \mathbf{X}_i + \mathbf{V}_i(1 - t)$        ▷ Estimated target particle $\hat{\mathbf{X}}_N \sim \mathbf{Z}_1$

$\quad\quad \mathbf{V}_i \leftarrow \mathbf{V}_i + \gamma(t) * \nabla_{\mathbf{X}_i} \mathcal{LL}(\hat{\mathbf{X}}_N)$          ▷ DiverseFlow step

$\quad\quad \nabla_{\text{MCG}} \leftarrow \frac{\partial}{\partial \mathbf{X}_i} \|\mathbf{Y} \odot \mathbf{M} - \hat{\mathbf{X}}_N \odot \mathbf{M}\|_2^2$     ▷ Manifold Constrained Gradient

$\quad\quad \mathbf{X}_{i+1} \leftarrow \mathbf{X}_i + \mathbf{V}_i \Delta_t$            ▷ Euler step

$\quad\quad \mathbf{X}'_{i+1} \leftarrow \mathbf{X}_{i+1} - \alpha_{t_i} \nabla_{\text{MCG}}$    ▷ Apply MCG correction; $\alpha_{t_i} = \sqrt{1 - t_i}$

$\quad\quad \mathbf{Y}_{i+1} \leftarrow \mathbf{X}_0(1 - t') + \mathbf{Y}t'$    ▷ Linearly interpolate between $\mathbf{X}_0$ and $\mathbf{Y}$ at $t_{i+1}$

$\quad\quad \mathbf{X}''_{i+1} \leftarrow \mathbf{X}'_{i+1} \odot (1 - \mathbf{M}) + \mathbf{Y}_{i+1} \odot \mathbf{M}$      ▷ Replace known region with $\mathbf{Y}_{i+1}$

$\quad$**end for**

$\quad$**return** $\mathbf{X}_N$

---

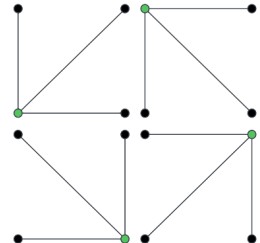 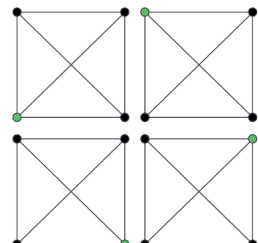

(a) Computing the sum of pairwise distances from current particle (green) to every other particle (black)

(b) Computing the determinant needs to consider the distance of every point from every other point

Figure 14: A geometric look at Particle Guidance and DiverseFlow

## D LIMITATIONS

From a modeling perspective, while DiverseFlow seeks to enhance the sample diversity of flow-matching models under a fixed sampling budget, it is still limited by the distribution modes the underlying FM models have learned. For instance, the word "mouse" may refer to: (i) a mammal (rodent), (ii) a computer peripheral. DiverseFlow could not generate any samples of the computer mouse with just the prompt "a mouse"; we hypothesize that the learned likelihood of the animal significantly dominates the latter meaning. Again, with SD-v3, we could not find any examples of coins for "a mint" which we could find for SD-v1.5. Thus, the discovery of diverse modes is still clearly dependent on the model being used.

From a computational perspective, for high-resolution generative modeling, estimating the diversity gradient $\nabla_{\mathbf{x}_t} \mathcal{LL}$ can be memory intensive. With either Stable Diffusion or LFM, it is necessary to backpropagate over (i) the KL-regularized AutoEncoder, (ii) the feature encoding ViT, $F$, and (iii) the high-resolution sample $\hat{x}_1$—thus practically limiting us to a batch of 4 samples at a time. We note that Particle Guidance faces a similar challenge. However, this can be overcome by computing a progressively growing kernel: we can sample a set of 4 images, and then sample another 4, where the kernel is $8 \times 8$, and another 4, where the kernel is $12 \times 12$, and so on, till some maximum allowed context or kernel size.

