# OpenReview forum: "DiverseFlow: Sample-Efficient Diverse Mode Coverage in Flows"
_ICLR.cc/2025/Conference — ICLR 2025 Conference Withdrawn Submission_

### Official Review · Reviewer_VAiX · 2024-10-29

**Soundness:** 3
**Presentation:** 4
**Contribution:** 2
**Rating:** 5
**Confidence:** 3

**Summary:**

This paper designed DiverseFlow, a training-free sampling method that increases output diversity in flow models without additional model evaluations, using determinantal point processes to enhance mode coverage.

**Strengths:**

- The proposed algorithm demonstrates a notable increase in sample diversity, effectively supporting the authors' initial claims about its ability to enhance mode coverage in flow generative models.
- The paper presents both controlled experiments in toy settings and evaluations in real-world image generation scenarios, providing a comprehensive analysis of the method's performance across different contexts.

**Weaknesses:**

- Section 6.4 uses abbreviations like CFM, SB-CFM, and OT-CFM without defining them right away, which leaves readers guessing. Clear definitions upon first mention would help make the section easier to follow.
- The image generation results mostly rely on displaying images to demonstrate diversity, with only a few qualitative measures tucked away in the appendix.
- The method's complexity—specifically, the cost of calculating gradients of eigenvalues at $O(k^3)$—could potentially be reduced to $O(k^2)$ with some algorithms, but it is still significant. This makes it hard to scale to larger values of sampling points $k$. For smaller values of $k$ (like $k=2$ to $10$, as mentioned in the paper), simply sampling multiple times independently could offer similar diversity without requiring much computation. Given recent advances in diffusion sampling techniques, normal sampling is not a significant issue.
- Adding extra terms to the sampling process can often make samples diverge from stable paths, which might be why the method still requires a high number of sampling steps (up to 100) to stabilize the sampling. This supports my previous point that it might be better to use accelerated sampling techniques and sample more points. The paper does address this.

**Questions:**

- The paper does not mention the role of classifier-free guidance (from Ho & Salimans, 2022) in influencing the diversity of sampling. How significant do you believe this factor is in relation to your proposed method?
- While the technical contributions of your work are noteworthy, I find it challenging to identify practical applications for the diversity sampling problem beyond those examples provided in Section 3. Can you elaborate on the demand for addressing this problem?

---

### Official Review · Reviewer_wKmw · 2024-10-30

**Soundness:** 3
**Presentation:** 3
**Contribution:** 2
**Rating:** 5
**Confidence:** 4

**Summary:**

The paper "DiverseFlow: Sample-Efficient Diverse Mode Coverage in Flows" deals with the problem of generating a more diverse set of generations in Flow models. To this end, a detrimantal point process is introduced as a coupling term into the ODEs that describe the Flow dynamics. Qualitative results are presented for text-guided image generation of polysemous words, image inpainting and class conditional image synthesis, and the model is shown to produce more diverse images. Quantitative superiority is also demonstrated against IID sampling, and Particle Guidance, on a synthetic mixute of Gaussian dataset.

**Strengths:**

+ the paper tackles a very interesting problem: how to generate diverse samples from a model, without needing excessive sampling
+ the proposed methodology, marrying DPPs with Flow models, makes sense and is sound
+ the paper is written clearly
+ the experiments show improvements against the chosen baselines

**Weaknesses:**

- while the theory seems compelling to me, the visual results seems to have a few issues, which should be mentioned and discussed in the paper: firstly, quality of sampled Diverseflow images seem to be worse - e.g. in Fig3(b lower image) the boxer seems to merge with the rope, or in Fig3(d) the lower white crane has a very unrealistic neck. Secondly, while the model adds diversity, it seems to also mix concepts for polysemous prompts: "a famous boxer" shows an image with a boxer (sports) and also a dog in the same image instead of either. Same for the mechanical crane, which has a shape akin to the animal, and also for the deer buck coin. Also, while Fig4 shows some changes even in identity (4c), the added diversity in Fig4e is very subtle to notice.
- In section 6.3. it is claimed that the model "maintain(s) the high quality of the samples and simultaneously explore more modes in the dataset." and then references Fig5. However, the diversity samples without CFG is arguably largest. This raises the question if there is maybe a CFG strength that leads to both diverse and high-quality samples. A plot that shows this front with varying CFG strength (e.g. Aesthetic score or FID on one axis, and diversity on the other) and demonstrating that the model really is outside this front, would put more compelling evidence behind the statement.
- a Limitation section is missing. Clearly the model has limitations. E.g. it would be important to discuss that the model needs more memory (when naively implemented at least), as the coupling between ODEs implies the need to keep all generations in memory simultaneously.
- the statement "observe that though DiverseFlow obtains better diversity at similar quality, the aesthetic score of Particle Guidance is quite low, suggesting poor
quality." should be made more clear. I.e. it's a bit unclear what 'better' refers to (presumably the IID model), and it should be pointed out that according to the plot PG has higher diversity, but this comes at the cost of lower aesthetic score. To make a more direct comparative statement between PG & DF it'd be great to push PG to higher aesthetic scores, or DF to more diversity (such that they have common aesthetic score values).
-  a bit more minor: ideally, add more trials in Figure 9b, such that DiverseFlow is not at 10, but rather 9.96 or something, to allow some future comparisons.

**Questions:**

- how much are the examples shown selected / what was the selection procedure? This should also be mentioned (e.g in Section 6.1, but also elsewhere)
- how much more memory is needed for generation due to the DPP coupling dynamics? Does this have implications for the resolution of the generated images?
- in Figure 8: why are both Particle Guidance and DiverseFlow changing the original duplicate image into the same modified image? This seems unlikely for independent models to me.

---

### Official Review · Reviewer_8rTs · 2024-10-31

**Soundness:** 3
**Presentation:** 3
**Contribution:** 2
**Rating:** 5
**Confidence:** 4

**Summary:**

The approach introduces a training-free sampling algorithm that produces a diverse set of samples in the target distribution via imposing a gradient constraint on the estimated velocity. Although prior methods may achieve generation diversity through repeated sampling until satisfactory outputs are obtained, this process is inefficiently costly and does not always guarantee good coverage of data modes. Specifically, this paper introduces a ODE-based sampling scheme with fixed sampling iterations K by first generating K target examples and computing the gradient of a determinant loss function w.r.t those examples (inspired from quantum physics). The method demonstrate the performance in enriching diversity of generated samples on text-to-image generation, class-conditional generation, and large-hole inpainting.

**Strengths:**

- The method proposes a fixed budget sampling scheme where only a certain number of examples are generated that covers sufficient modes of target data.
- The main idea is derived from quantum physics, offering a theoretical guarantee.
- Limitation is thoroughly mentioned in the appendix.

**Weaknesses:**

- Although the method is theoretically sound, its performance and evaluation remain questionable. Need to provide more quantitative metrics like precision & recall [1].
- I’m curious whether using variance as a simple gradient function is still effective. Since the core idea of Determinantal Point Processes (DPP) is to promote diversity in generated samples through the feature space of a pretrained model, variance could also serve as a useful indicator for this purpose. This can be considered as a simple baseline to compare with.
- DPP bears some similarities with PCA (principle component analysis), particularly in choosing the most K principle eigen components to better represent the original space. The authors should expand on this comparison.
- Is the imposed gradient constraint applicable during the training phase? Because the method is inadvertently limited to the coverage modes of pretrained model.
- Figure 5, using DiverseFlow seems to increase the contrast and make the color saturated. Why is that?
- The guidance scale in CFG sampling is known to negatively impact the diversity of generated examples. In figure 5, the method used a scale of 8 which is really large. Why do the authors not use a lower scale like under 3.0 for experiments? The gain of method is really marginal in this experiment.
- Honestly, the improvements shown in qualitative Figures 3, 4, 5, and 8 are not very convincing, as the differences are minimal in some cases.




Paper exposition should be improved:
- The captions of figure 2 have rendering issues. Should add notation of green dots in the figure for clarity.
- The notation of source and target distribution are unclear as in figure 2a.
- How $\gamma_t$ is constructed?

Ref:
[1] Kynkäänniemi, Tuomas, et al. "Improved precision and recall metric for assessing generative models." Advances in neural information processing systems 32 (2019).

**Questions:**

Please address my concerns above.

---

### Official Review · Reviewer_mnNK · 2024-10-31

**Soundness:** 4
**Presentation:** 4
**Contribution:** 3
**Rating:** 8
**Confidence:** 3

**Summary:**

I like the niche problem selected by the authors on improving diversity in sampling in generative models, particularly the category normalizing flow.
Well-defined questions with an extended set of experiments do show some promising results.
I'd be happy to see future work continuing to other types of generative models and minimizing the limitations mentioned by the authors.

**Strengths:**

What worked for me:
1. The narrowing down of a problem to focus on a particular generative model within a sampling budget for text, image modality
2. The computational experiments on the right dataset that impact the results
3. A well-written paper supporting the correct theory, background, and mathematical equations

**Weaknesses:**

1. Authors would have shown how DiverseFlow would work for non-polysemous prompts, and/or for sentence-wide prompts "A boxer is running on a trail" vs "A boxer"?
2. Authors have shown the diversity through 4x4 grid however it will be informative to know the diversity proportion in the sampling budget.
experiment concerning sampling budget
3. Could you add how other generative models behave in the absence of such a diverse sampling scope?
4. Re-check the 6.5 section for the quoted results in bold. Maybe, check all other sections too for any missing equations or typos.

**Questions:**

I am not an expert in this area, so would need to have more opinions. Also, if the authors want to make any corrections or add anything that I am missing in my review, please let us know.
1. The sampling budget or its measurement unit is not clearly defined
2. Figure 2 captions are not well aligned and thus hard to understand
2. LINE #344 CFG's abbreviation is missing

---

### Note · Authors · 2024-11-12

**Comment:**

We thank the reviewers for their valuable and insightful feedback. At this time, we would like to withdraw the paper.

**Withdrawal Confirmation:**

I have read and agree with the venue's withdrawal policy on behalf of myself and my co-authors.